# High-Speed Infrared Thermography for Measuring Flash
# Temperatures in Sheared Fault Gouge Analogues
Chien-Cheng Hung[1, 2, 3] and André R. Niemeijer[1]
*[1]Department of Earth Sciences, Utrecht University, Utrecht, The Netherlands*
*[2]Department of Earth Sciences, National Central University, Taoyuan, Taiwan*
*[3]Institute of Earth Sciences, Academia Sinica, Taipei, Taiwan*
*Corresponding author: Chien-Cheng Hung (e-mail: c.hung@uu.nl)*
**Abstract**
Flash temperatures induced by flash heating can lead to thermal softening or
decomposition of fault-zone materials at microscopic grain contacts and, consequently,
cause a rapid reduction in fault strength during seismic slip. To quantify the efficiency
of short-term frictional heating at the contact scales and its impact on the mechanical
fault strength, we conducted rotary-shear friction experiments on Ottawa quartz sand
"gouges" with variable grain sizes of 250–710 $\mu$m at a range of normal stresses of 1–
7.5 MPa and slip velocities of 1–50 mm/s under room-dry and wet conditions. We
employed a high-speed infrared camera to monitor temperature fluctuations along the
outer circumference of the ring-shaped gouge layer during sliding, utilizing a frame rate
of up to 1200 Hz with a spatial resolution of 15 $\mu$m to capture flash temperature
occurring at asperity contacts. We show that flash temperature can be captured within
the gouge layer in both room-dry and wet conditions with a peak value up to ~220°C
and ~100°C, respectively. In addition, the flash temperature increases with increasing
slip velocity and grain size, while decreasing at higher normal stress, which is likely
associated with enhanced grain size reduction. In our study, we showed that flash
temperatures in shearing fault gouges can be constrained using a fast thermal camera.
Although difficulties remain in the experimental set-up related to the need to confine
the gouge layer and to the evolution of contact size due grain size reductions, the trends
in maximum temperatures we observed agree with those predicted from theory.
**Short summary**
During coseismic slip, rapid frictional sliding along the fault surface generates heat,
triggering thermally activated processes that soften fault materials and reduce fault
strength. Understanding heat generation and temperature rise during slip is key to
characterizing these processes. Flash heating at highly stressed contacts is considered
one of the most common dynamic weakening mechanisms for both bare rock and fault
gouge at the onset of rapid slip. However, experimental constraints on flash
temperatures in sheared granular gouge are limited. In this study, we developed an
experimental setup with a high-speed infrared camera to capture in-situ thermal images
across the full thickness of a gouge layer during rapid shearing under dry and wet
conditions. Our goal is to experimentally determine how peak flash temperature
depends on loading and material conditions, and to test whether the results align with
predictions from theoretical models.

**1. Introduction**
Characterizing temperature rise from frictional sliding is crucial to understanding
the thermally activated mechanisms that cause faults to weaken during coseismic slip
and associated earthquake rupture propagation (Rice, 2006). Several theoretical and
experimental studies have proposed various dynamic weakening mechanisms,
depending on fault rock composition and experimental conditions, to account for the
reduction in friction coefficient seen when sliding velocities approach seismic slip rates
($V > 0.01$ m/s; see Di Toro et al., 2010, Niemeijer et al., 2012, and Tullis, 2015 for
summaries). Among these weakening mechanisms, flash heating at highly stressed
asperity contacts is believed to be one of the main mechanisms causing dramatic
weakening at the onset of rock-on-rock slip (e.g. Goldsby and Tullis, 2011; Kohli et al.,
2011; Spagnuolo et al., 2015) and shearing of gouge materials (e.g. Yao et al., 2016a,
b). So far, the inferred role of flash heating and weakening is generally based on
predictions using a flash-heating model (Beeler et al., 2008; Brantut and Platt, 2017;
Proctor et al., 2014; Rice, 2006; Yao et al., 2018) and/or the microstructural evidence
of thermal softening (e.g. local melting) at asperity contact scales (e.g. Acosta et al.,
2018). Recent studies using laboratory friction experiments by Saber (2017) and
Barbery et al. (2021, 2023) have shown that surface temperature increases at contact
points on sliding surfaces between Westerly granite blocks, under room-dry conditions,
can be successfully captured by employing a high-speed infrared camera (a frame rate
of 300 Hz and a resolution of 75 $\mu$m). These authors documented that flash temperatures
and stress distribution are highly heterogeneous at the rock-on-rock interface during
seismic slip. In addition, no evidence of dynamic weakening was detected, as the
measured surface temperatures did not approach the weakening temperature inferred
for granite (~800°C). This may also reflect wear processes that reduce local normal
stresses and increase the true contact area during slip, thereby suppressing conditions
favorable for flash heating. Currently, experimental constraints on flash temperatures
in a sheared granular gouge layer are lacking. In particular, direct detection of thermal
signals, from flash heating occurring within a shearing gouge layer, remains
challenging due to difficulties with gouge confinement, the small scale of grain contact
asperities (<<1 mm), and short contact lifetimes (ms).

In an attempt to make new progress, we developed an experimental setup, with a
high-speed infrared camera, that allows us to acquire in-situ thermal images across the
entire thickness of the gouge layer during rapid shearing under room-dry and wet
conditions. In this study, we aim to experimentally characterize the dependence of peak
flash temperature on normal stress, slip velocity, and grain size to better understand
how flash heating is influenced by these variables, and to assess whether the results are
consistent with the predictions of previous theoretical analyses (Rice, 2006; Proctor et
al., 2014; Brantut and Platt, 2017).

**2. Materials and Methods**
**2.1 Starting materials**
We used ASTM standard C778 Ottawa quartz sand obtained from the U.S. Silica
Company (Ottawa, IL, USA) as starting materials. The materials contain a $SiO_2$ content
of 99.7 wt%, minor quantities of $Al_2O_3$ (0.06 wt%), $Fe_2O_3$ (0.02 wt%), and $TiO_2$ (0.01
wt%). Scanning electron microscopy showed that most sand grains are well-rounded,
with smooth surfaces characterized by pressure solution indentations and quartz
overgrowths (Hangx and Brantut, 2019). We prepared four fractions of the sand with
grain sizes of 250–300 $\mu$m, 300–425 $\mu$m, 425–500 $\mu$m, and 500–710 $\mu$m by double-
sieving the material as received. We adopted the grain size fraction 425–500 $\mu$m as the
reference grain size for most of the experiments. For each experiment, we used 35
grams of the Ottawa sand gouge, which results in an initial gouge thickness of ~4.5 mm
after normal loading at 2.5 MPa (effective) normal stress and before shearing.

**2.2 Rotary-Shear Friction Experiments**
We performed medium-velocity friction experiments using a rotary-shear
apparatus (RAP; Korkolis, 2019) plus a high-speed infrared camera (Figure 1a). The
RAP consists of a torque reaction frame that is housed inside an Instron 8862 testing
machine equipped with a servo-controlled electromechanical actuator that may be
operated either in position control (±50 mm range, 5 μm resolution) or in load control
mode (±100 kN range, 0.008 kN resolution). Within the RAP, we developed a gouge
setup that allows us to confine gouge samples and pressurize the pore fluid within the
simulated fault zone during rapid sliding (Figure 1b). The gouge layer is confined
between two ring-shaped, steel pistons (100-mm external, 70-mm internal diameters),
allowing loading in the axial direction. The piston faces are toothed, with the teeth
spaced at 1.5 degree (240 teeth on the full ring) and their height varies from the inside
diameter of 0.62 mm to the outside diameter of 0.96 mm. In the middle, the height is
0.79 mm. All teeth are oriented perpendicular to the direction of sliding. A brass outer
and inner rings were used to confine the gouge layer laterally to avoid gouge extrusion
during shearing. The driving platen is equipped with two angular potentiometers (0.001
degrees, or about 0.74 μm resolution) that measure its rotation. A pair of load cells (20
kN range, 0.008 kN resolution), mounted on opposite sides of a horizontal steel block
("crosshead"; Figure 1a), measure the reaction force of the frame due to the rotation
imposed by the motor. Axial displacment is tracked by measuring the change in the load
frame position under load control. Local pore fluid pressure changes during sliding
were measured using two pressure transducers (10 MPa full range, with 0.01 MPa
resolution and less than 10 ms response time) installed at opposite sides, and at identical
radial position (centered at $d$ = 85 mm), in the bottom rotary piston (Figure 1b). To
detect thermal signals across the full gouge layer thickness, we installed a circular,
transparent sapphire window (10 mm in diameter and 2 mm in thickness) in the outer
confining ring (Figure 1c, d). The screw-in design of the window allows its position to
be adjusted to be as close as possible to (in contact with) the gouge layer.

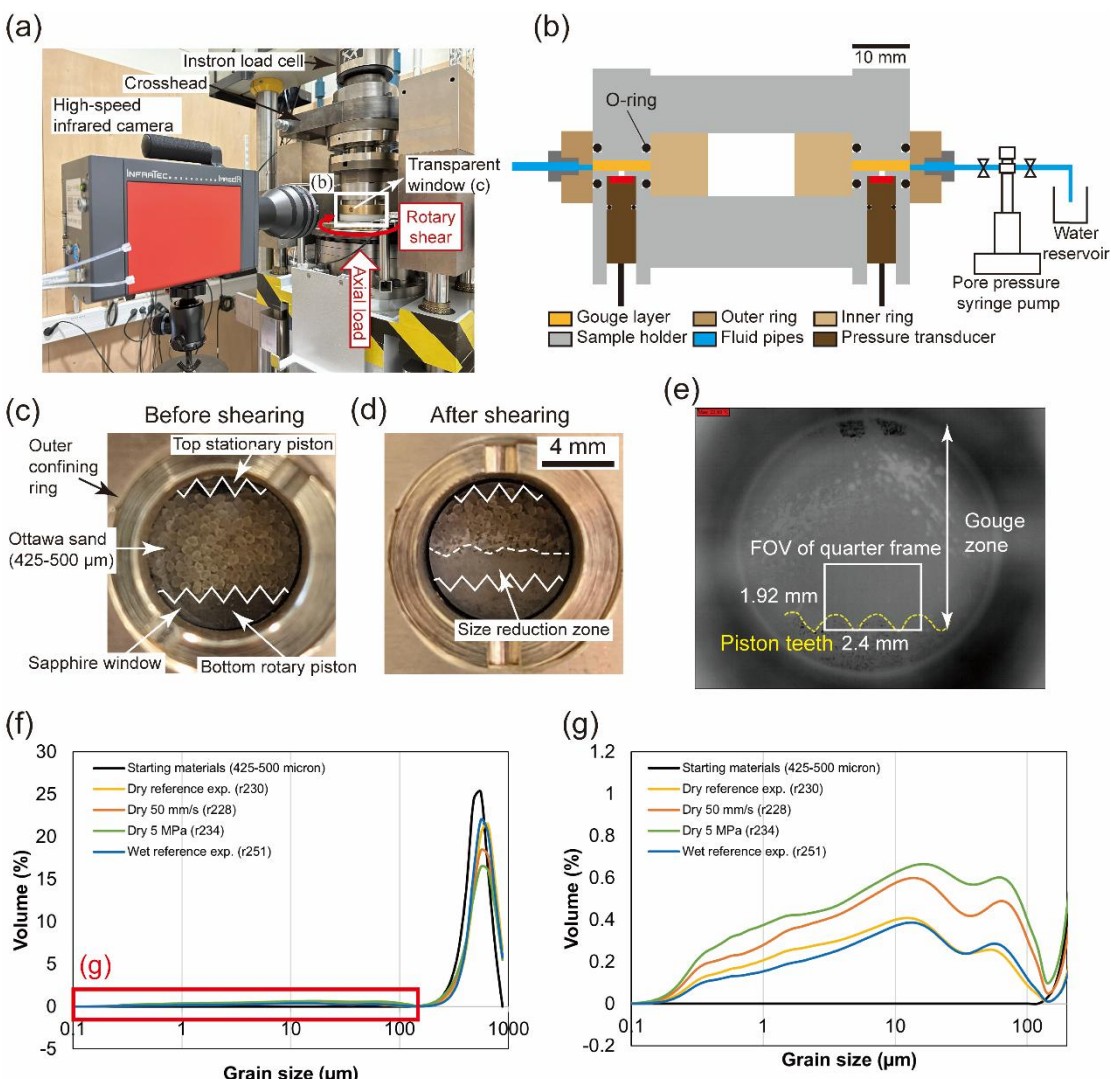

**Figure 1.** Experimental setup. (a) Photograph of the pressurized gouge setup installed
in the rotary-shear apparatus (RAP) with the high-speed infrared camera. (b) Cross-
section of the pressurized gouge setup. Pressure transducers are installed in the bottom
rotary piston at ~2 mm distance from the gouge layer. The pore fluid inlet and outlet
are at 135 and 45 degrees to the sapphire window, respectively. (c) Photograph of the
profile of the undeformed, room-dry Ottawa sand gouge layer, sandwiched between the
pistons, and laterally confined by an outer brass ring with a transparent sapphire
window. (d) Photograph of a deformed room-dry gouge layer showing a zone/layer of
reduced grains. (e) Thermography of the gouge layer before sliding under a full-frame
field of view (FOV) (9.6 mm × 7.68 mm). The white rectangle indicates the field of
view (FOV: 2.4 mm × 1.92 mm) under quarter-frame analyses (1200 Hz) applied in all
our tests. (f) Particle size distribution (PSD) of the starting materials (425-500 $\mu$m
Ottawa sand gouge) and the deformed samples with the same initial grain size under
different conditions, plotted as volume percentage vs. grain size. Reference experiment
refers to a normal stress of 2.5 MPa and slip velocity of 10 mm/s (g) Zoom-in PSD for
grain sizes 0.1–100 $\mu$m.

Friction experiments were conducted on both room-dry and wet gouges at an
applied normal stress $\sigma_n$ of 1.0–7.5 MPa and a constant slip velocity $V$ in the range 1–
50 mm/s, reaching a total displacement of up to 750 mm (equivalent to ~2.8 of full
rotations of the piston assembly). Grain comminution was lowest in our experiments at
lower normal stresses and sliding velocities, as shown in the particle size distribution
anaylsis (Figure 1f, g). Thus, we chose a normal stress of 2.5 MPa and a slip velocity
of 10 mm/s as the reference conditions at which flash heating could be resolved spatially.
Before applying the normal load to the target value, wet gouges were prepared by
saturating the sample with DI water using a syringe pump (ISCO pump; Figure 1b)
through the pore fluid inlet port until water came out of the outlet drainage ports,
without bubbles. All wet experiments were conducted under undrained conditions at
initially atmospheric pore pressure conditions (i.e. 0.1 MPa) by keeping the outlet ports
closed during shearing. After the target normal stress was applied, we initiated shearing
of the gouge sample, recording the thermal imaging at synchronized time. During the
entire experiment, data on shear stress $\tau$, normal stress $\sigma_n$, axial displacement, and
velocity $V$, were obtained at a logging rate of 5 kHz. All the tests performed are listed
in Table 1.

**Table 1.** *List of experiments and conditions with the average temperature increase $\Delta T_{avg}$*
*as well as the measured and predicted (see equation 1) flash temperatures $T_{max}$. The IR*
*temperature range chosen for the room-dry and wet experiments is 60–200°C and 30–*
*150°C, respectively. We define a flash as a maximum temperature measurement in a*
*single frame that is larger (>50°C for dry, >4°C for wet) than the maximum*
*temperatures in neighbouring frames. Predictions of flash temperature for wet*
*conditions are not included.*

| ID | $\sigma_n$ (MPa) | Pore fluid | $V$ (mm/s) | $D$ (mm) | Grain size (mm) | Number of flashes | $\Delta T_{avg}$ (°C) | Measured $T_{max}$ (°C) | Predicted $T_{max}$ (°C) |
|---|---|---|---|---|---|---|---|---|---|
| | | | | | *Room-dry conditions* | | | | |
| r228 | 2.5 | - | 50 | 500 | 0.425-0.5 | 30 | 61.84 | 183.54 | 568.95 |
| r229 | 2.5 | - | 25 | 750 | 0.425-0.5 | 31 | 55.87 | 222.60 | 408.17 |
| r230* | 2.5 | - | 10 | 750 | 0.425-0.5 | 23 | 48.42 | 137.72 | 265.50 |
| r231 | 2.5 | - | 1 | 75 | 0.425-0.5 | N/A | 12.42 | N/A | 97.63 |
| r232 | 2.5 | - | 5 | 600 | 0.425-0.5 | 23 | 38.82 | 124.25 | 193.59 |
| r234 | 5 | - | 10 | 750 | 0.425-0.5 | 28 | 50.92 | N/A | 291.41 |
| r235 | 1 | - | 10 | 400 | 0.425-0.5 | 27 | 36.44 | 123.96 | 228.63 |
| r236 | 7.5 | - | 10 | 200 | 0.425-0.5 | 1 | 52.37 | 90.26 | 309.32 |
| r237 | 2 | - | 10 | 750 | 0.425-0.5 | 15 | 49.68 | 221.98 | 251.46 |
| r238 | 3 | - | 10 | 750 | 0.425-0.5 | 22 | 45 | 169.36 | 272.23 |
| r239 | 7.5 | - | 50 | 450 | 0.425-0.5 | 1 | 73.24 | 101.34 | 666.94 |
| r240 | 2.5 | - | 10 | 750 | 0.3-0.425 | 23 | 53.62 | 125.88 | 228.63 |
| r241 | 2.5 | - | 10 | 750 | 0.25-0.3 | 2 | 52.69 | 119.03 | 202.98 |

| | | | | | | | | | |
|---|---|---|---|---|---|---|---|---|---|
| r242 | 2.5 | - | 50 | 750 | 0.25-0.3 | 4 | 60.83 | 150.07 | 429.16 |
| r245 | 2.5 | - | 10 | 750 | 0.5-0.71 | 30 | 53.33 | 222.60 | 297.51 |
| *Wet conditions* | | | | | | | | | |
| r249 | 2.5 | DI water | 50 | 750 | 0.425-0.5 | 3 | 8.74 | 81.74 | - |
| r251* | 2.5 | DI water | 10 | 750 | 0.425-0.5 | 13 | 5.28 | 43.74 | - |
| r254 | 5 | DI water | 10 | 750 | 0.425-0.5 | 27 | 4.69 | 158.40 | - |
| r255 | 5 | DI water | 50 | 750 | 0.425-0.5 | 43 | 7.32 | 61.67 | - |
| r256 | 7.5 | DI water | 10 | 750 | 0.425-0.5 | 15 | 5.22 | 85.42 | - |
| r257 | 4 | DI water | 10 | 750 | 0.425-0.5 | 6 | 3.77 | 44.17 | - |
| r258 | 6 | DI water | 10 | 750 | 0.425-0.5 | 4 | 3.18 | 38.40 | - |
| r259 | 2.5 | DI water | 10 | 750 | 0.5-0.71 | 5 | 3.85 | 40.09 | - |
| r260 | 2.5 | DI water | 10 | 750 | 0.25-0.3 | *N/A* | 3.49 | *N/A* | - |

*Note. N/A denotes no flashes observed. The symbol "*" indicates reference experiments*
*for the room-dry and wet conditions. $\sigma_n$ = the applied normal stress. V = the imposed*
*slip velocity. D = the imposed total displacement.*

**2.3 High-Speed Infrared Thermal Imaging of Gouge Profiles**
We used an Infratec ImageIR 8300hp infrared camera with a precision
microscope lens M=1.0x and a working distance of 20 cm (Figure 1a) to provide the
requirements for high-speed thermography. The combined setup has a detector format
of 640 × 512 pixels and a wide range of IR frame capture rates from up to 355 Hz (a
full frame window size) to up to 5000 Hz (a line measurement) with a spatial resolution
down to 15 $\mu$m per pixel. All the infrared radiation (IR) signals acquired from the
infrared camera were processed using IRBIS® 3.1 Infrared Thermographic Software
(https://www.infratec.eu/thermography/thermographic-software/) to convert the digital
IR signal data to thermographs (i.e. temperature distributions) with the user choice
calibration (e.g. emission, absorption, and transmission). In this study, we took an
emissivity of 1 (full reflection of IR from the object), an absorption coefficient of 0 (no
attenuation of IR), and a transmission of 1 (full penetration of incident IR) as the inputs
for the calibration of the IR signal to arrive at the target object temperature in the

software. We have not taken the effects of mineralogy, the sapphire window or the presence of water on the IR signal into account in the present work. The focus of the lens was adjusted until the piston teeth were visible in the gouge layer on the real-time thermograph (Figure 1e). During shearing of each experiment, we adopted a quarter-frame analysis of up to 1200 Hz (~0.83 ms) with a field of view (FOV) of 2.4 mm × 1.92 mm and a spatial resolution of 15 $\mu$m per pixel to capture thermographs that emerged from the FOV of the gouge shear zone. These mostly covered the bottom third of the gouge layer (i.e. rotating side) plus a small portion of the piston teeth (Figure 1c, e). For experiments conducted at the chosen reference conditions (ID r230, r251), a sequence of at least 90000 thermographs (75 seconds × 1200 Hz) was acquired from the FOV throughout the experiment. The IR camera is equipped with a neutral density filter and is capable of measurements over several pre-set, calibrated temperature ranges with multiple integration times. We took the factory-calibrated settings for two different IR temperature ranges of 60–200°C and 30–150°C, for room-dry and wet test conditions, respectively. After each experiment, we exported the maximum temperatures and the corresponding location (X and Y coordinates), as well as the average temperatures over the FOV of every acquired thermograph, to investigate the temporal and spatial evolution of peak temperatures with displacement.

## 3. Results

### 3.1 Mechanical Behavior

Figure 2 shows the evolution of apparent friction $\mu_{app}$ (= shear stress $\tau$ / applied normal stress $\sigma_n$) against displacement obtained for the room-dry and wet gouges at the variable slip velocities, normal stresses, and initial grain sizes shown. For the room-dry gouges (Figure 2a, c, e), the frictional evolution is generally similar, regardless of slip

velocity, normal stress, and grain size. Specifically, $\mu_{app}$ firstly overcomes a "static"
peak friction $\mu_s$ of ~0.9, soon after slip initiation, subsequently rapidly drops to a
constant (dynamic) level of ~0.65 within ~0.3 m displacement. This is followed by
minor displacement weakening beyond the first full rotation (~267 mm displacement)
reaching a near-constant value of ~ 0.6 during the late stage of slip. The overall level
of friction systematically decreases with increasing normal stress (Figure 2c), with the
exception of the experiment conducted at 5 MPa normal stress (ID r234), which shows
anomalous behavior in the form of marked displacement strengthening after one
rotation. This is likely due to jamming of material between the piston and the confining
ring.

226        For the wet gouges (Figure 2b, d, f, and 3), the initial frictional behavior is similar

to the room-dry gouges, featuring a peak $\mu_{app}$ value $\mu_s$ of ~0.9 followed by a sharp
friction drop. Different levels of slip weakening occur depending on the slip velocity,
normal stress, and grain size. At a normal stress of 2.5 MPa, the wet gouge subjected to
a slip velocity of 50 mm/s showed increased slip-weakening towards a lower dynamic
friction of ~0.45, compared with the behavior seen at the lower slip velocity of 10 mm/s
(Figure 2b) or in dry samples. At a slip velocity of 10 mm/s, the wet gouges significantly
slip-weaken only when the normal stress is equal to or greater than 4 MPa. Under these
conditions, the onset of weakening starts earlier and the dynamic friction level
decreases with increasing normal stress (Figure 2d). Increased slip weakening is also
observed, compared with the reference test, when the grain size is either greater or
smaller than the reference value, without any systematic correlation between grain size
and the level of dynamic friction (Figure 2f).

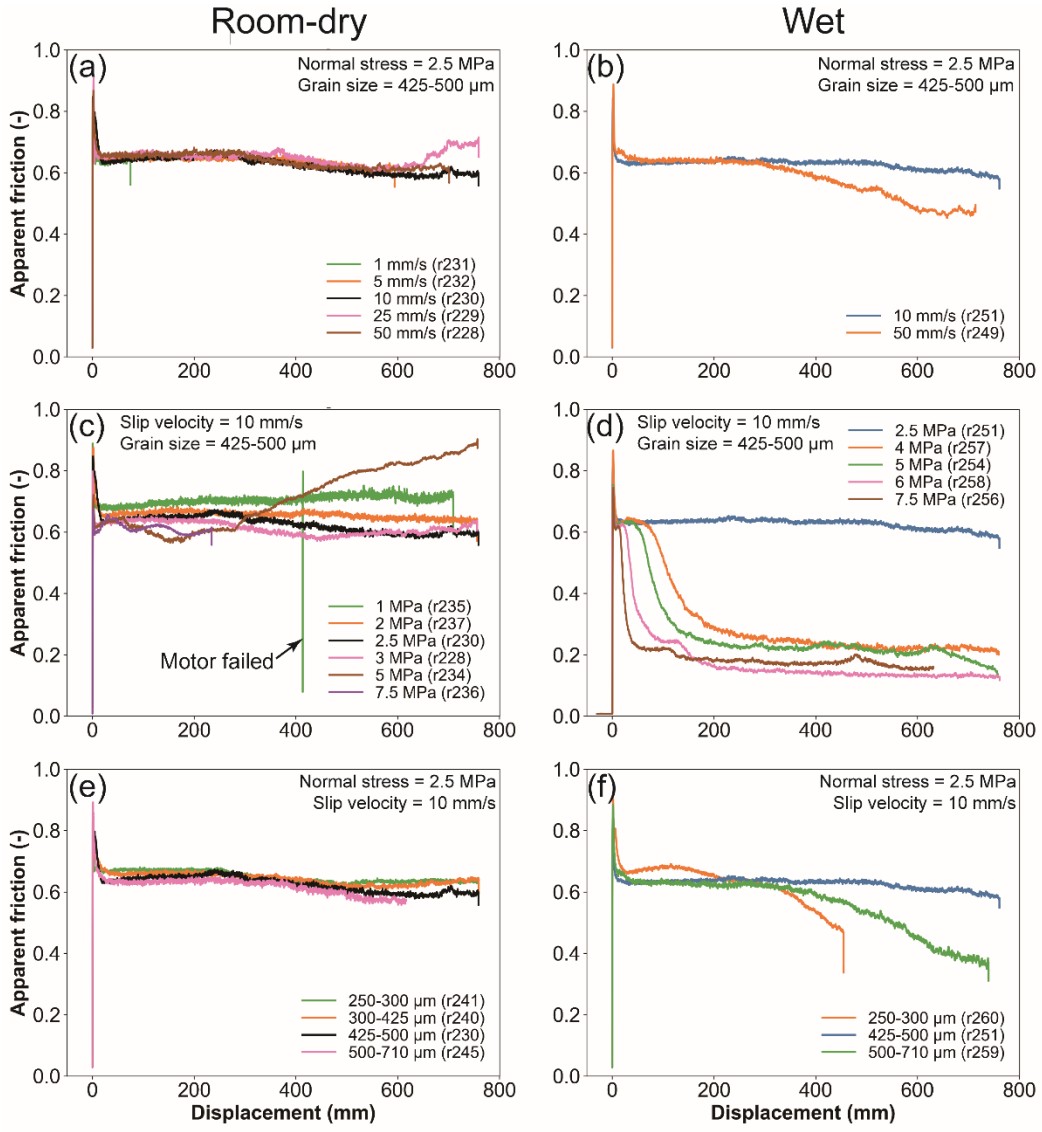

**Figure 2.** Evolution of apparent friction $\mu_{app}$ as a function of displacement for room-dry (a, c, e) and wet gouges (b, d, f) under variable slip velocities, normal stresses, and initial grain sizes. The sudden friction drop observed in (c), the green curve, is the result of re-shear due to a motor failure.

Fluid pressure increases were always observed in the wet experiments at the displacement at which weakening starts (i.e. beyond the peak static friction; see Figure 3). This shows that the fluid pressure increase has a positive correlation with applied normal stress, since the maximum fluid pressure increased from ~3.5 MPa, at a normal

stress of 4 MPa, to ~7 MPa at a normal stress of 7.5 MPa. In addition, each experiment shows a sudden gouge dilatation during the attainment of peak friction, followed by ongoing gouge compaction (Figure 3). The amount of compaction shows the same trend with normal stress as fluid pressure, i.e. more compaction occurs in experiments with higher normal stress.

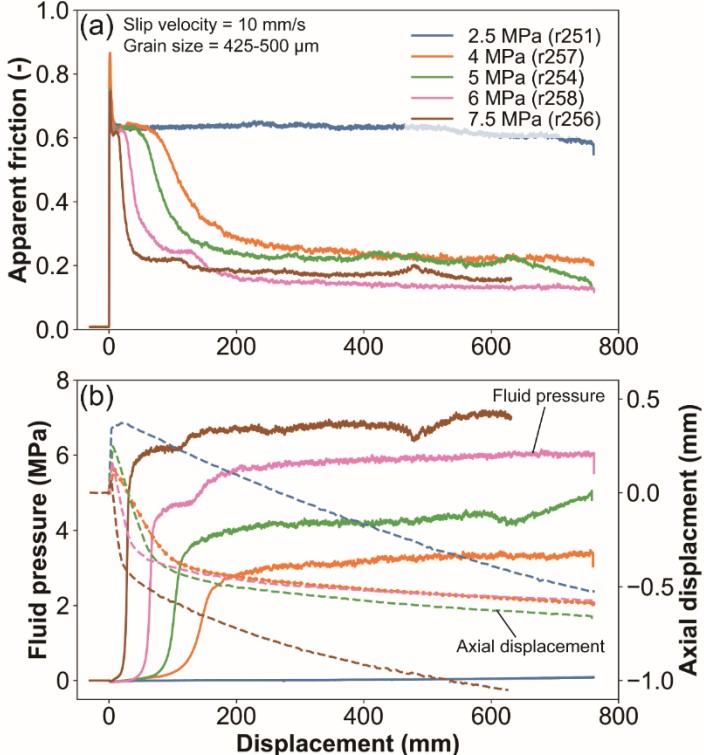

**Figure 3.** Evolution of (a) apparent friction and (b) pore fluid pressure and axial displacement, as a function of shear displacement for the wet experiments at a slip velocity of 10 mm/s, using an initial grain size of 425–500 µm. Dashed curves in (b) represent axial displacement data whereby positive values indicate dilatation and negative values indicate compaction. Solid curves in (b) represent pore fluid pressure in these undrained tests.

**3.2 Evolution of Maximum Temperature Rise with Displacement**

The maximum temperatures obtained from each thermal image are plotted as a
function of displacement in Figures 4, 5, and 6. Generally, multiple temperature peaks
(i.e. spikes) with a duration of one frame (~0.83 ms) can be observed, representing
flashes in most experiments with the largest peak temperatures being ~220°C and
~100°C in the room-dry and wet gouges, respectively. For most of the experiments, the
maximum temperatures in the FOV showed a rapid increase after the slip initiation,
with a faster rise with increasing velocity, but then soon reached a plateau (at a base
level of ~25–40°C), which becomes higher at higher velocity (Figure 4). The presence
of a temperature plateau is due to the settings of the IR temperature measurement range
(60–200°C and 30–150°C for room-dry and wet experiments, respectively); below the
lower bound of the range, temperature cannot be accurately determined or displayed.
Another thing that should be noted is that Figures 4, 5, and 6 include all thermal flashes
that occurred within the full FOV throughout the entire slip period of each experiment.
Thus, the observed flashes might not only result from the sliding between grain-grain
contacts but also from sliding between the grains and the window as well as between
the grains and the piston, due to gouge extrusion or grain fragments jamming between
the sapphire window and the piston.

Nonetheless, these results still provide insight into systematic differences in the
evolution of temperature and the magnitude of the peak temperatures. Overall, we
observed that peak temperature rises are significantly reduced in the presence of water,
and the peak temperature systematically increases with increasing slip velocity (Figure
4). In both the room-dry and wet gouges, the number of flashes becomes much less and
their magnitudes become much lower at the lowest applied slip velocity of 1 mm/s. In
the dry experiments, the magnitude of the flashes appears to be the greatest (up to 200°C)
at 2.5 MPa, while becoming less significant with both decreasing and increasing normal
stress (Figure 5a). The dependence of peak temperature on normal stress is not visible
in the wet experiments (Figure 5b). Note that the dry experiment (ID r236) at 7.5 MPa
normal stress was terminated soon (~3 seconds) after slip initiation (Figure 5a) because
of sounds indicating unwanted abrasion of the outer ring. For the range of grain sizes
explored, the variations in the magnitudes of flashes are similar in both the dry and wet
tests and are difficult to quantify based on these data (Figure 6).

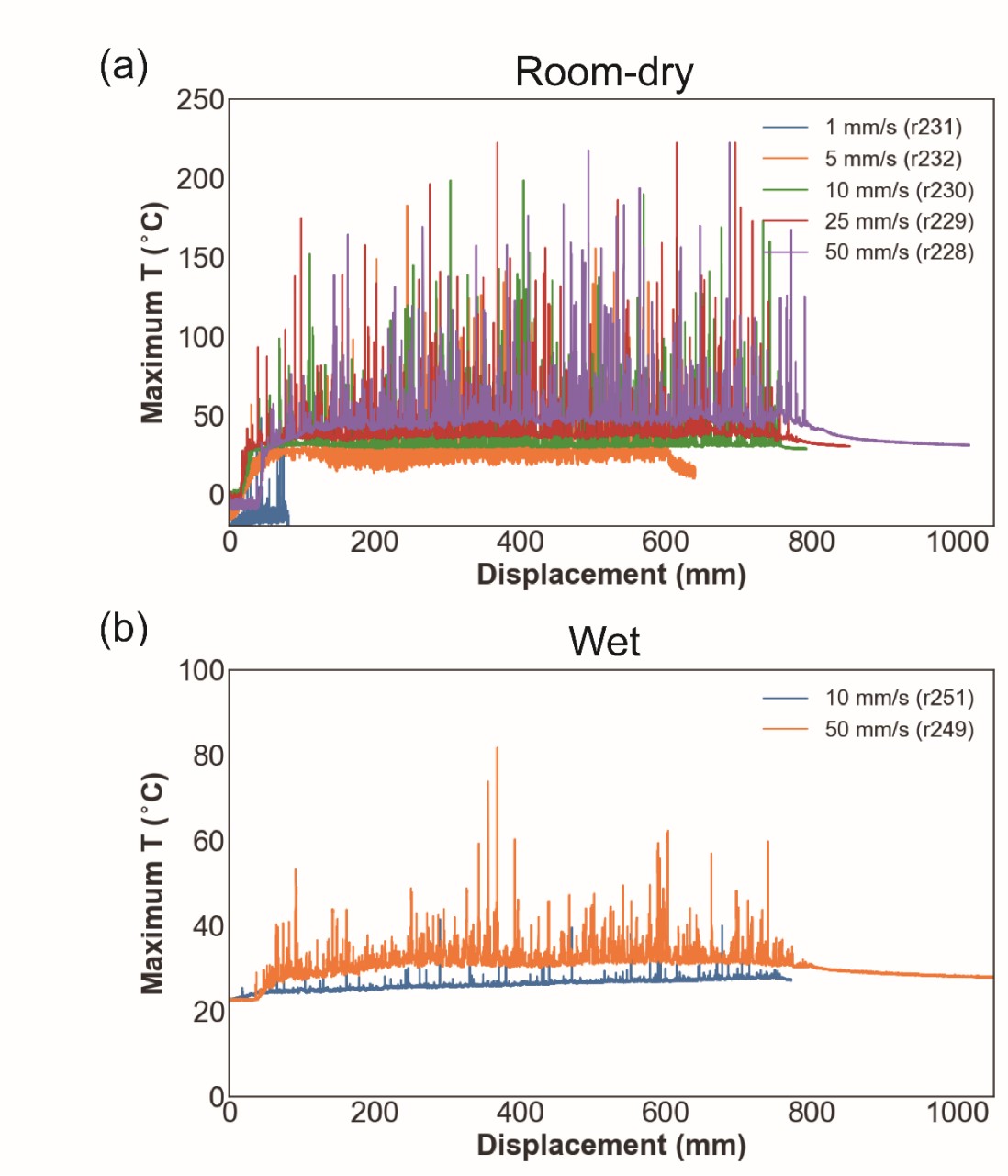


**Figure 4.** Evolution of maximum temperature within the FOV (4 mm × 3.2 mm) as a

function of displacement for the room-dry gouges (a) and the wet gouges (b) tested at

a normal stress of 2.5 MPa and different slip velocities.


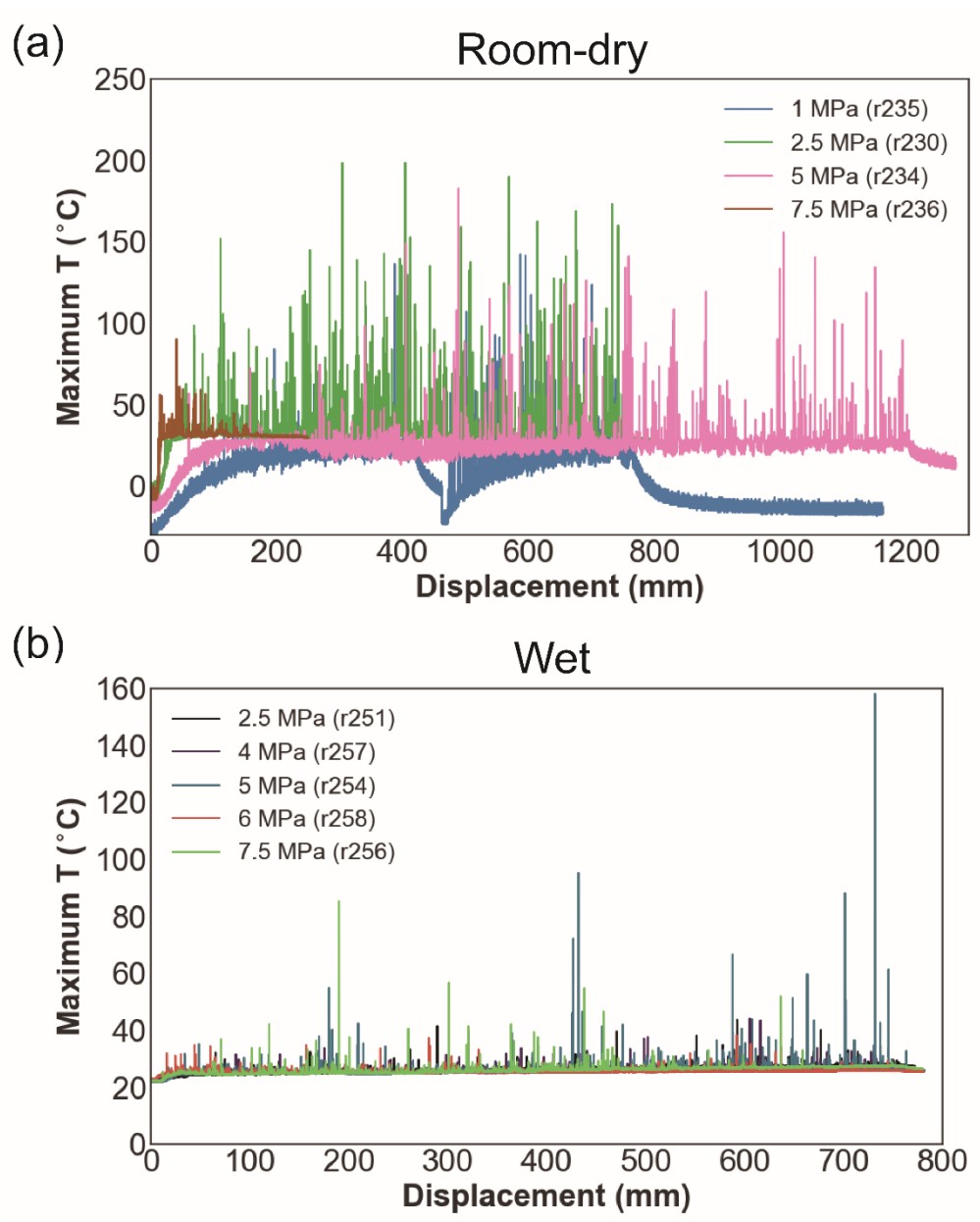


**Figure 5.** Evolution of maximum temperature within the FOV (4 mm × 3.2 mm) as a
function of displacement for the room-dry gouges (a) and the wet gouges (b) tested at
a slip velocity of 10 mm/s and different normal stresses. The temporary decrease in the
minimum measured temperature at normal stress of 1 MPa results from an unintended
stop in shearing at ~40 seconds due to a motor break.

308

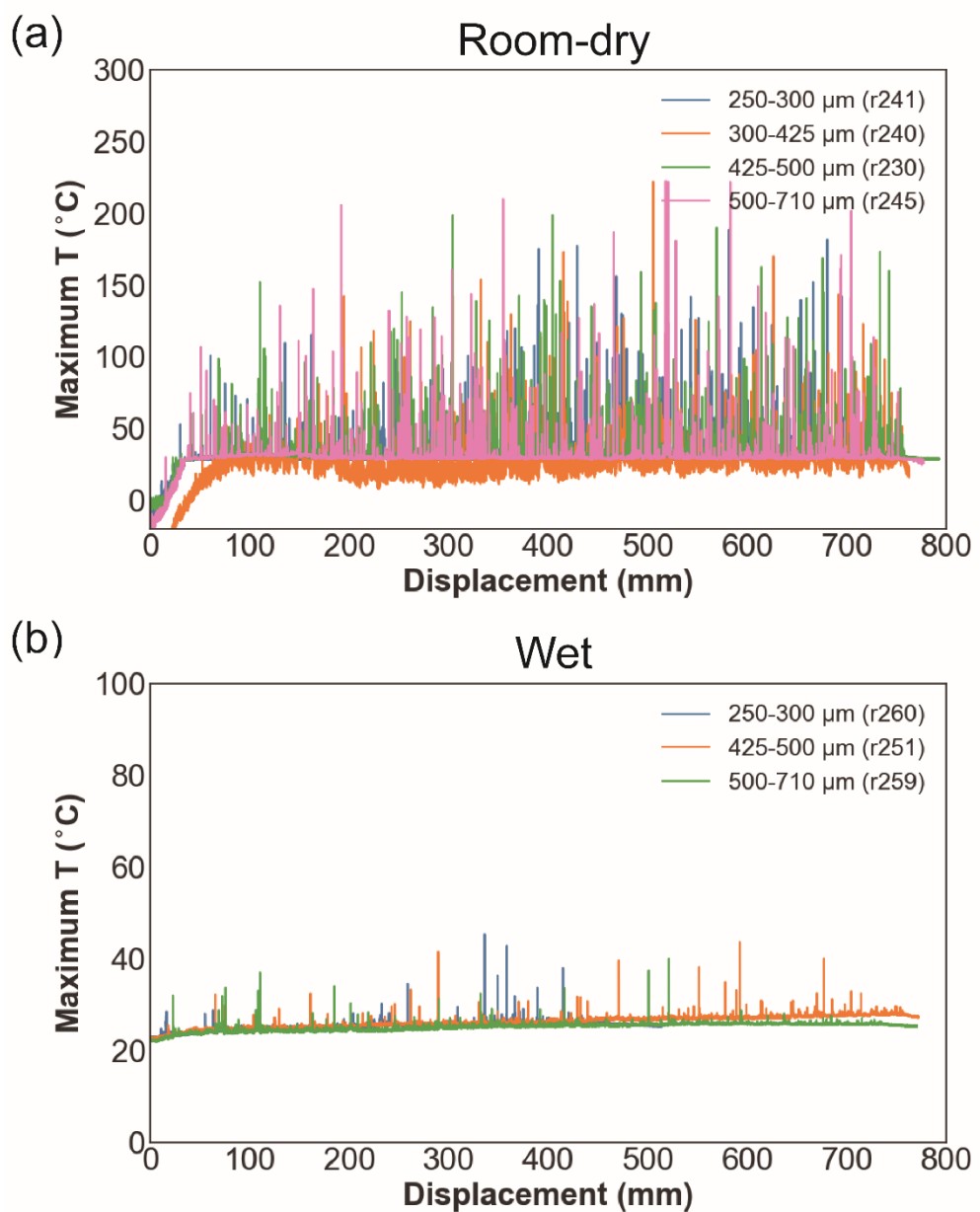

**Figure 6.** Evolution of maximum temperature within the FOV (4 mm × 3.2 mm) as a function of displacement for the room-dry gouges (a) and the wet gouges (b) tested at a normal stress of 2.5 MPa and a slip velocity of 10 mm/s, with different initial grain sizes.

**3.3 Spatial and Temporal Distributions of Flashes**

We located the position of the thermal flashes to see if they occurred within the actual gouge zone, as opposed to the portion of the FOV below the teeth tops. Figure 7

presents the spatial and temporal distributions of the flashes, here taken as those frames
(duration ~0.83 ms) with temperature rise $\Delta T$ larger than 50°C and 4°C, for the
reference room-dry and wet gouges, respectively. These results exclude the flashes with
a position at and below the horizontal level (i.e. Y position) of the teeth tops on the
bottom rotary piston (dashed lines). This is to rule out the flashes that potentially
resulted from sliding between the grains and the piston / window. After filtering out
data from below the chosen moving datum, we still observed many flashes, which we
infer occurred within the main gouge zone. The peak temperature and the number of
these flashes in each experiment are summarized in Table 1. Generally, in terms of
spatial distribution, flashes tend to be concentrated within a narrow zone (~500 $\mu$m
thick) under room-dry conditions (Figure 7a, b) whereas flashes are more separated
under wet conditions.


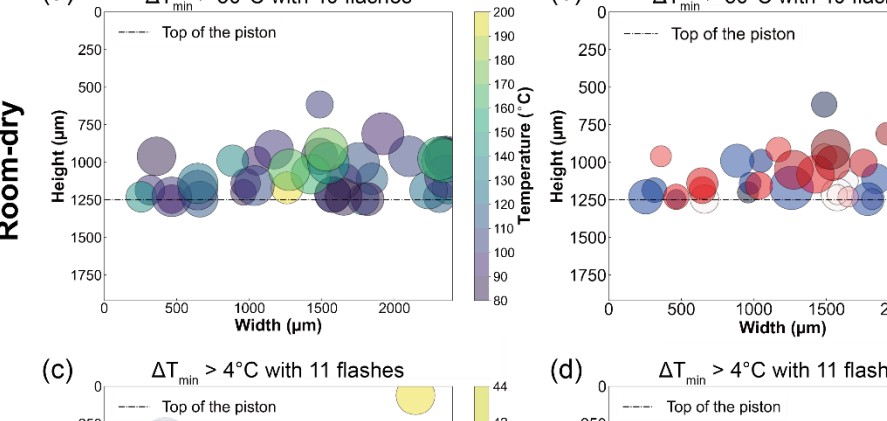
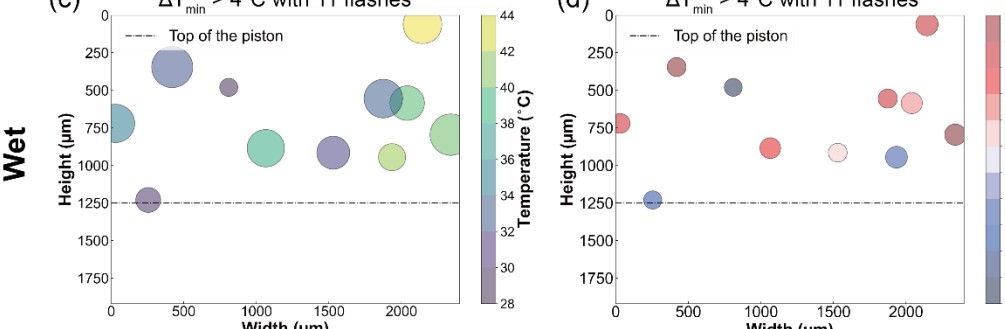


**Figure 7.** Spatial and temporal distribution of flashes within the main gouge zone with
a minimum temperature rise $\Delta T$ (relative to the bulk maximum temperature) larger than
50°C and 4°C for the reference experiments of the room-dry (a and b; ID r230 with 40
flashes) and wet gouges (c and d; ID r251 with 13 flashes), respectively. Tests
performed at 2.5 MPa normal stress and 10 mm/s sliding velocity. Larger dots indicate
later occurrence (a, c) or higher temperature rise (b, d).

**3.4 Estimation of Flash Temperature**
Based on the flash-weakening model (e.g. Rice, 2006; Beeler et al., 2008; Proctor
et al., 2014; Yao et al., 2018), the flash temperature on an asperity (grain) contact ($T_a$)
within a gouge layer can be expressed as:

$$T_a = T_{bulk} + \frac{\tau_c}{\rho c} \sqrt{\frac{(V/N_a)D_a}{\pi\alpha_{th}}} \qquad (1)$$

Here, $T_{bulk}$ is the initial bulk temperature of the fault zone, $\tau_c$ is the contact shear
strength, $\rho$ is the gouge density, and $c$ is the specific heat capacity of the gouge, $N_a$ is
the number of asperity contacts sharing the total slip velocity $V$, $D_a$ is the diameter of
the asperity contact, and $\alpha_{th}= K/\rho c$ is the thermal diffusivity of the gouge material,
where $K$ is the thermal conductivity. For a first-order estimate, the latent heat of possible
reaction (e.g. fluid evaporation) is not considered in the equation. Here we only applied
the model to the dry samples. For the dry Ottawa quartz sand with a grain size of 425–
500 $\mu$m, we took $\tau_c$ as ~9–12 GPa (Goldsby and Tullis, 2011), $\rho$ as 2650 kg/m3, $c$ as
730 J/kgK and $\alpha_{th}$ as ~1.5 × $10^{-6}$ m²/s. Considering the thickness of the shear zone in
dry gouges (~4 mm or ~16–20 grains), where ~50% of the gouge layer was involved in
shearing (Figure 1d), this results in $N_a$ values ranging from ~8 to 10, which is within a
reasonable range, comparable to previous studies (4–10 Yao et al., 2018; 10–20 Rice,
2006; 4–13 Proctor et al., 2014; 10 Chen et al., 2023). For $D_a$, we assumed two different
particle packings (hexagonal packing vs. centered cubic packing) to estimate the range
of $D_a$ between two spherical grains, assuming Hertzian contact mechanics. Based on
the elastic moduli of Ottawa sand (e.g. Erdoğan et al., 2017), for the grain size of 450–
500 μm, we adopt average values of $D_a$ as 13, 18, 22, and 25 $\mu$m at normal stresses of
1, 2.5, 5, 7.5 MPa, respectively. At a normal stress of 2.5 MPa, we take $D_a$ to be 10, 13,
17, and 23 $\mu$m for grain sizes of 250–300 $\mu$m, 300–425 $\mu$m, 450–500 $\mu$m, and 500–710
$\mu$m, respectively.

Figure 8 shows the comparison of the maximum flash temperature obtained from
the model prediction and from the measurements for the dry experiments. Overall, the
measured values are lower than the predicted ones. Predicted flash temperature
increases with increasing slip velocity, normal stress, and grain size, following equation
1, but this increasing trend is less (or not) visible in the measurement. The discrepancy
between the predicted and measured flash temperature tends to increase with slip
velocity and normal stress (Figure 8a, b) but remains relatively consistent with variable
grain sizes (Figure 8c).

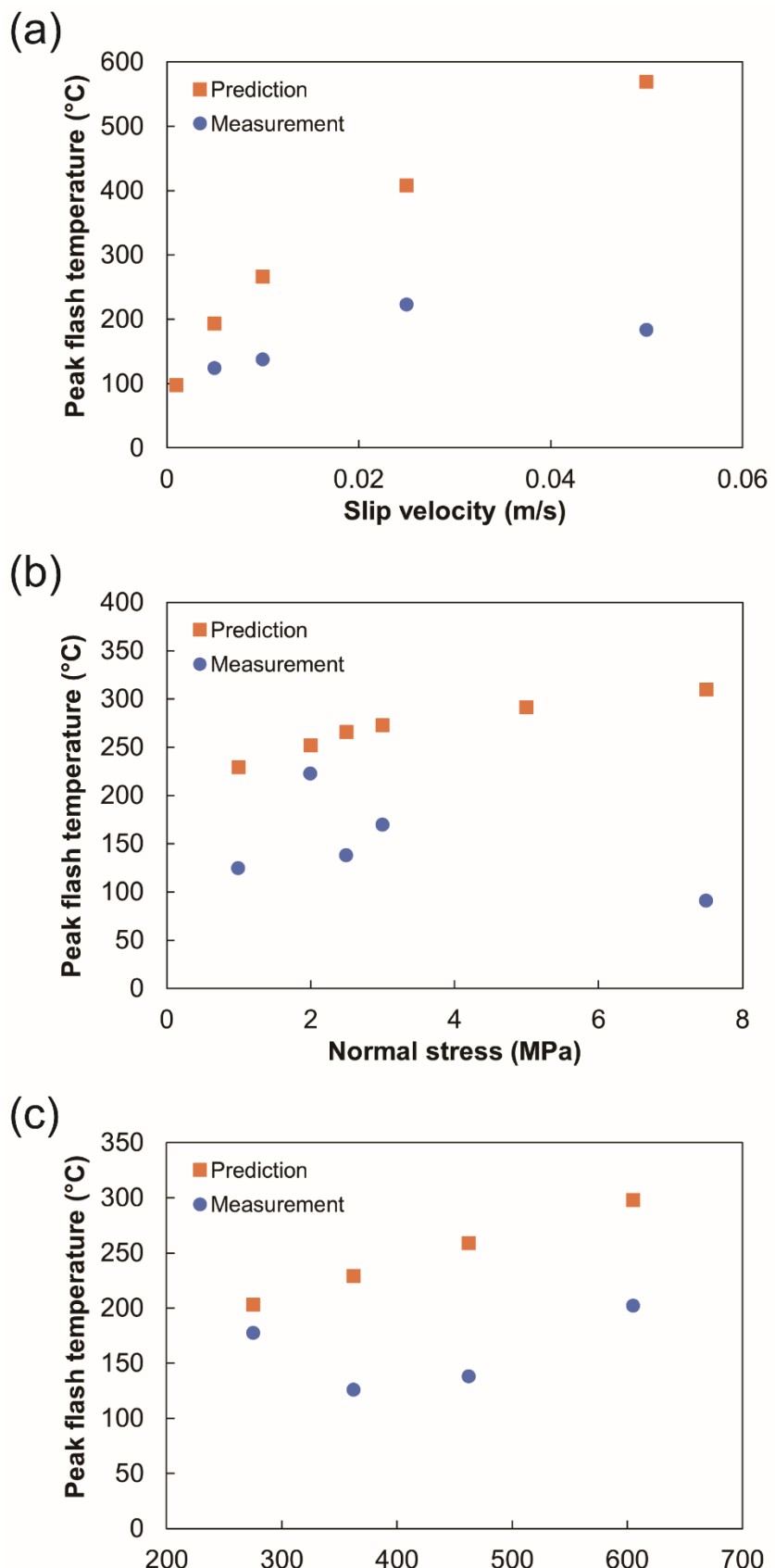


**Figure 8.** Comparison of the peak flash temperatures between the prediction based on
the flash weakening model (Rice, 2006) and the infrared measurement with different
slip velocities (a), normal stresses (b), and grain sizes (c). The measured peak flash
temperature at a slip velocity of 1 mm/s is not shown in (a) because the overall measured
temperatures are lower than the selected temperature range (60–200°C).

**4. Discussion**
**4.1 Robustness of the Flash Temperature Measurement**
In the literature, flash heating is suggested to occur within a short contact lifetime
(ms), depending on the scale of asperity contact (typically $\mu$m) as well as the sliding
velocity at the contact (Rice, 2006). The current setting of the infrared camera can
accurately measure thermal information with spatial and temporal resolutions of 15 $\mu$m
and ~0.83 ms, respectively. This means that the measured temperature might be
underestimated if the contact diameter is smaller than 15 $\mu$m or if the contact lifetime
is shorter than 0.83 ms (e.g. Madding et al., 2007). Another thing to be noted is that
even if the grains are perfectly aligned with the window, the contact would still be a
distance away from the window, assuming spherical grains. That means that the camera
is probably not in focus for the contacts themselves, which would also underestimate
the measured temperature. Our measurement of the maximum temperature within the
FOV show that multiple peak temperatures (i.e. flashes) with duration of one frame (i.e.
0.83 ms) were recorded from the observed region of both the room-dry and wet gouges
when the imposed slip velocity is higher than 1 mm/s (Figure 4).

Figure 9 shows the contact lifetime against the contact diameter $D_a$, assuming a
specific grain size and slip velocity. The $D_a$ is estimated following the method
described in section 3.4. The contact lifetime is then obtained from the ratio of $D_a$ to
the contact sliding velocity (= imposed slip velocity $V$ / number of asperity contacts
$N_a$). Based on the calculated contact diameter of the asperity $D_a$ (10–25 $\mu$m) and the
assumed number of asperity contacts $N_a$ (10), the contact lifetime of the asperity for
our samples and experimental conditions should range from 10–25 ms to 0.2–0.5 ms at
the imposed sliding velocity of 1 mm/s to 50 mm/s, respectively. These ranges lie
around the limit of the temporal resolution of the camera, though the contact lifetime
for $V$ larger than 5 mm/s appears to be shorter than the duration of one frame, being
strongly controlled by the estimated values of the $D_a$ and $N_a$. This implies that the
measured flashes likely resulted from flash heating between grains. Here, we should
note that most of the calculated asperity sizes are smaller than the spatial resolution of
the camera, suggesting that the actual temperature might be higher than the
measurement. Moreover, if the flashes have a shorter duration than the frame duration,
their actual temperature was also likely higher than the measured values.

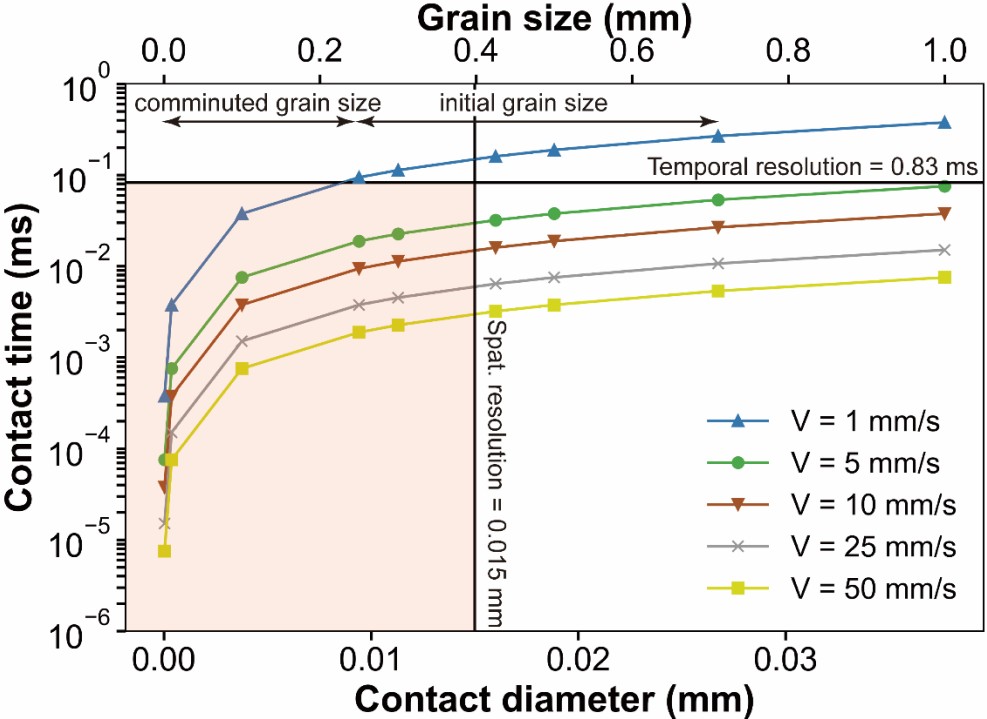


**Figure 9.** Contact lifetime against contact diameter $D_a$ for flashes for a specific grain size and slip velocity. The $D_a$ is estimated following the method described in section 3.4. $N_a$ is assumed to be 10. The black horizontal line gives a temporal resolution of 0.83 ms, and the vertical line has a spatial resolution of 0.015 mm. The light red rectangle region represents the region for which the IR camera with the current settings have neither sufficient spatial nor temporal resolution.

After filtering out the flashes occurring within the region of the rotating piston, we could still observe a few tens of flashes within the actual gouge zones under both room-dry and wet conditions (Figure 7; Table 1). Among these flashes, we cannot exclude the possibility that some flashes result from the sliding of a grain against the window due to the lateral force and the dragging shear force acting on the grains that are attached to the window. Thus, these flashes might exhibit different characteristics in terms of shape in the thermographs due to the potential of scraping of the grain along the sapphire

window. Figure 10 shows three different characteristics of flashes that can be observed
from the thermographs. Generally, most of the flashes have a circular shape with the
highest temperature at the center which gradually decreases towards the boundary
(Figure 10a, b). Some of these flashes are located within the interior of grains (Figure
9a), where the individual grain outline can be identified, suggesting the flashes might
result from grain-to-window sliding. For other flashes, it is difficult to identify their
position relative to the solid grains (Figure 10b). Yet another type of flash displays a
tail shape (Figure 10c) which may indicate the existence of grain-to-window flash. With
the current data, it is difficult to distinguish whether the flashes were actually generated
at grain-to-grain contacts or whether they were mainly related to grain-to-window
sliding. To avoid the grain-to-window flashes, improvements in the experimental set-
up are required, such as using a curved window to help ensure that the grains can pass
along the window without creating temperature flashes. Another possibility might be to
include an IR-transparent low-friction coating on the window.

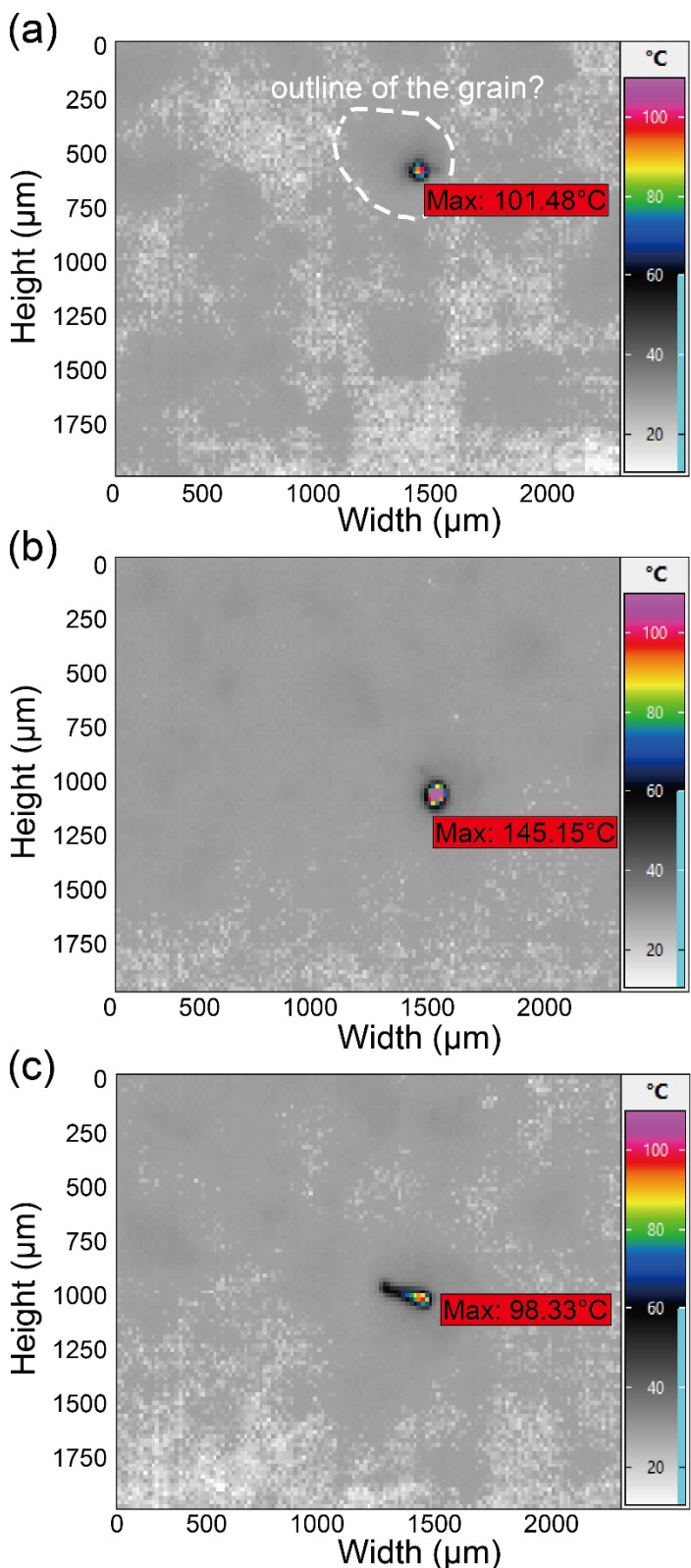

**Figure 10.** Thermographs with a flash identified in the actual gouge zone, as obtained from the reference room-dry experiment (r230) at displacements of 293 mm (a), 676 mm (b), and 526 mm (c).

Comparing our measurements with model predictions shows that measured flash temperatures are consistently lower than predicted, and that the discrepancy increases with normal stress and slip velocity (Figure 8). As discussed above, the limited spatial and temporal resolution of the IR camera may contribute to these differences by underestimating the true peak temperatures. Another possible explanation is that the IR camera only captures temperatures at the flanks of the gouge layer or grain contacts, rather than at their centers. Because stress concentrations at contact asperities generate extremely steep temperature gradients, even a small offset from the central portion of the contact results in significantly lower measured temperatures. This geometric constraint likely contributes to the observed discrepancy between measured and predicted flash temperatures. We argue that the increasing discrepancies could be related to the process of grain size reduction during frictional sliding, and/or possibly the setting of the IR temperature ranges. As can be seen in the measured particle size distribution of the deformed sample (Figure 1g), abundant fine grains were generated during the experiment. The initial grain size (425–500 $\mu$m) is reduced to a minimum value of ~50 nm, with the largest proportion of grains smaller than the starting material having a size of 10 $\mu$m. In addition, the proportions of fine grains are systematically larger at a higher normal stress and slip velocity. This indicates that high normal stress and slip velocity increase the degree of grain size reduction, which leads to a decrease in asperity size $D_a$ and a possible increase in $N_a$ (at a constant thickness of the localized zone) as well as an increased contribution of fracture energy from grain comminution. The decrease in asperity size would affect affect our ability to resolve flash temperatures in space and time, because smaller contacts persist for shorter durations (Figure 9). The

increase in fracture energy suggests a potential energy sink that is not converted into
heat, although we found that this contribution is negligible compared to the frictional
work. If grain size reduction is one of the reasons causing the increasing variations, this
implies that the measured flash temperatures should decrease with displacement due to
the limitation of the spatial resolution or the peak temperatures might move away from
the zone of grain size reduction. However, we do not observe these behaviors. To
characterize the flash temperature distribution in the gouge layer robustly, multiple
temperature ranges or a combined temperature range should be utilized in multiple
repeat experiments (Barbery et al., 2021). Despite all this, our results still show some
dependence of flash temperature on slip velocity and grain size.

**4.2 Mechanical Behaviors under Room-Dry and Wet Conditions**
We observed that the measured average temperature and flash temperatures can
reach up to ~73°C and ~220°C, respectively, under room-dry conditions (Table 1).
Despite high flash temperatures, suggesting the high effectiveness of flash heating, the
room-dry gouges do not show any significant weakening which remains at a constant
friction level of 0.6 (typical quasi-steady friction for quartz; Byerlee, 1978). This might
be explained by the relatively low measured and predicted flash temperatures as
compared to the softening (i.e. melting) temperature of quartz (~1720°C, Spray, 1992).

By contrast, the water-saturated gouges reveal slip weakening accompanied by
fluid pressure increase at relatively high normal stresses (>2.5 MPa) and slip velocities
(>10 mm/s), while the average and flash temperature rises are only up to ~9°C and
~80°C, respectively (Figure 4, 5, and 6; Table 1). Firstly, this suggests that temperature
increase can be strongly limited by the presence of water, though the effect of water on
the IR signal has not been considered in the data processing. Secondly, the above
temperature increases of 9°C and 80°C are insufficient to account for the observed fluid
pressure increase (3.5–7 MPa). We expect the bulk fluid pressure increase to be less
than 1 MPa, taking a reasonable range for the thermal pressurization factor $\Lambda$ (e.g.
0.075–0.11 MPa/°C for quartz-rich gouges; Hunfeld et al., 2021). On this basis, we
suggest that the mechanism causing slip weakening in the wet samples is most likely
associated with compaction-induced pressurization (compaction softening - Oohashi et
al., 2011), which is supported by the data shown in Figure 3.

At the same time, however, other candidate mechanisms related to the presence of
water, such as thermomechanical pore fluid pressurization (Rice, 2006; Badt et al.,
2020), flash pressurization resulting from local thermal pressurization at grain contacts
induced by flash heating (e.g. Chen et al., 2023; Yao et al., 2018; Hung et al., 2025),
and/or the water/vapor phase transition (e.g. Chen et al., 2017) cannot be completely
excluded. Silica-gel lubrication due to amorphization of silica and gel formation via
shearing and comminution in the presence of moisture (Di Toro et al., 2004; Goldsby
and Tullis, 2002) might be another possible candidate for the dynamic weakening seen
in fast shearing of wet quartz materials. To test these hypotheses, we need to ensure the
gouge sample is well-compacted before the initiation of a high slip velocity to prevent
significant compaction during the experiment. This is typically achieved by applying a
conditioning stage (i.e. shear at low slip velocity) to the gouge layer. However, this was
intentionally not included in the current study because the initial grain size reduction
and associated comminution products would limit thermal radiation from grain contacts.
In addition, thermal flashes would become difficult to measure due to the shorter
asperity contact lifetime.

**5. Conclusions and Future Work**
In this study, we perform rotary-shear friction experiment on sheared fault gouge
analogues with a high-speed infrared camera to characterize flash temperatures induced
by flash heating at grain contact scales and their dependence on normal stress, slip
velocity, and grain size. The important findings in the present work are as follows:
1.   We can successfully monitor and quantify flash heating with instantaneous
temperature rises occurring in an experimental fault gouge under room-dry and
wet conditions.
2.   The presence of pore water likely limits flash temperature rise, though the extent
of the effect is unclear due to uncertainties in the extent of IR absorption of thermal
signatures by the water phase.
3.   Flash heating can be effective in dry gouge at a medium velocity (5 cm/s) but is
insufficient to cause dynamic slip weakening in a dry gouge-filled fault because
the flash temperature is too low to reach the softening temperature of the gouge
grains.
4.   The slip weakening effects that we observed in our wet experiments but not in dry
tests are best explained by compaction softening, though a contribution from
thermal pressurization at some scale cannot be eliminated.
5.   We have experimentally demonstrated that the flash temperature depends on slip
velocity and grain size, to some extent, in broad agreement with theoretical
predictions.

Final, we listed future work to further improve our understanding of flash heating
in a sheared granular layer on the basis of the current results:
1. Using normal high-speed camera to observe the sliding velocity at grain-to-grain
contacts.
2. Investigating the dependence of peak flash temperature on other particle
parameters like grain shape (round vs. angular), grain roughness, as well as
different background fluid pressures and pore fluids.

**Data availability**
All raw data can be provided by the corresponding authors upon request.

**Author contribution**
Conceptualization: André R. Niemeijer; Data curation: Chien-Cheng Hung, André R.
Niemeijer; Formal analysis: Chien-Cheng Hung, André R. Niemeijer; Funding
acquisition: André R. Niemeijer; Investigation: Chien-Cheng Hung, André R.
Niemeijer; Methodology: Chien-Cheng Hung, André R. Niemeijer; Project
administration: Chien-Cheng Hung, André R. Niemeijer; Resources: André R.
Niemeijer; Supervision: André R. Niemeijer; Validation: Chien-Cheng Hung;
Visualization: Chien-Cheng Hung; Writing (original draft preparation): Chien-Cheng
Hung; Writing (review and editing): André R. Niemeijer

**Competing interests**:
The authors declare that they have no conflict of interest.

**Financial support**
This work is part of the research programme DeepNL, financed by the Dutch Research
Council (NWO); Grant: DEEP.NL.2018.040. Additional support for the purchase of the
high-speed infrared camera was provided by the Nederlandse Aardolie Maatschappij
(NAM).

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
