# Peer review of "High-Speed Infrared Thermography for Measuring Flash"

_EGUsphere, 2025_

## Author Response (AR1)

3[th]  December  2025

**Cover letter of manuscript egusphere-2025-4591 "High-Speed Infrared Thermography for Measuring Flash Temperatures in Sheared Fault Gouge Analogues"**

Dear Dr. Fedeli,

Thank you for the opportunity to improve our manuscript. We have addressed and responded to all of the reviewer's comments with the greatest care and made changes to the manuscript wherever possible based on every comment, which we believe addresses all points raised.

Below, you will find a point-by-point response to the reviewer's comments. The reviewers' comments are in black and our responses are in blue. In the revised manuscript, all changes to the text have been highlighted in blue. We hope that our revisions prove to be satisfactory, and we look forward to hearing from you.

Yours sincerely,

***Chien-Cheng Hung,*** on behalf of the authors

Reviewer #1: Lu Yao

This is a great paper reporting measurements of flash temperature in simulated quartz gouge using infrared thermography. I am very interested in this topic myself, and have also attempted to measure flash temperature as the authors have done. However, the infrared camera in our lab (InfraTec8800) has a very narrow factory-calibrated temperature range, which was optimized for studying the tempo-spatial evolution of the temperature field during earthquake nucleation in half-meter sample friction experiments. Due to technical restrictions, it is not easy to send the camera back to Germany for recalibration or adjustment. As a result, I eventually had to abandon the attempt. I am thus excited to read through the manuscript. The manuscript is logically structured, with accurate and precise descriptions and well-designed figures. While reading through the Results section, I had a few questions in mind; however, most of them were addressed or discussed later in the Discussion section.

We thank Dr. Yao for the positive and encouraging comments on our manuscript. We are pleased to hear that the topic and approach resonate with your own research experience, and we appreciate your feedback. We have addressed all comments in detail below and hope that the revisions satisfactorily resolve the issues raised.

My overall recommendation is thus minor revision, as I have only one major comment and a few minor comments detailed below.
My major comment concerns the authors' interpretation of the much lower flash temperatures measured in the experiments compared to those predicted by the model. The authors should note that, when measuring the flash temperature from the flank of the gouge layer or from the flank of two grains in contact, the central portion of the contact is not actually exposed to the infrared camera. Due to the large stress concentration at the contact, there is likely to be an extremely steep temperature gradient in its vicinity, so even a small distance from the central portion, the peripheral areas of the contact may have much lower temperatures than at the center. This important limitation should be taken into consideration when interpreting differences in flash temperatures between the experimental measurements and the model predictions.

We thank Dr. Yao for this constructive comment. We fully agree that the discrepancy between the measured and predicted flash temperatures may arise, at least in part, from the spatial limitations of the temperature measurements. As noted, the infrared camera does not capture the central portion of individual grain contacts, where the

highest stress concentrations, and therefore the highest temperatures, are expected. We have added associated discussion in the main text to (Lines 453-459):

*"Another possible explanation is that the IR camera only captures temperatures at the flanks of the gouge layer or grain contacts, rather than at their centers. Because stress concentrations at contact asperities generate extremely steep temperature gradients, even a small offset from the central portion of the contact results in significantly lower measured temperatures. This geometric constraint likely contributes to the observed discrepancy between measured and predicted flash temperatures."*

Two minor comments:
(1) Why was the highest slip velocity used in the experiments limited to 50 mm/s? Was this constraint due to the capabilities of the rotary-shear machine, or was it related to the allowable temperature measurement range of the infrared camera? At normal stresses of a few MPa, gouge layers may exhibit pronounced slip weakening at slip velocities exceeding several hundred mm/s. If an experiment reveals significant weakening concurrent with high flash temperatures (e.g., close to 1000 degC; I believe the weakening temperature $T_w$ could be much lower than the melting temperature of undeformed standard quartz grains), you can have a more in-depth discussion of the flash heating mechanism.

We thank Dr. Yao for this comment. The slip velocity of 50 mm/s was constrained by the mechanical limits of the RAP rotary-shear apparatus for the current sample geometry. The infrared camera was not the limiting factor, as it is capable of measuring much higher temperatures with Multi Integration Time (MIT), ranging from 10°C to 1500°C—with acquisition rates up to 1200 Hz, depending on the selected range. We agree that experiments conducted at higher slip velocities (>100 mm/s), where more pronounced flash weakening and elevated flash temperatures may be expected, would allow a more in-depth discussion of the flash heating and weakening mechanism. However, faster velocities would also reduce the lifetime of the contacts and a higher acquisition rate of thermal image would be needed. Future experiments in a different apparatus capable of higher velocities are needed to evaluate whether flash heating in gouges can be captured with our current system.

(2) Line 331: Just to confirm: larger dots indicate higher temperature rise in a & c, and later occurrence in b & d, right? Also, is the unit of time on the Y-axis of panels b and d correct?

We thank Dr. Yao for checking these details. Yes, the first interpretation is correct. We have corrected this in the main text. Regarding the second question, the time units on the Y-axis of panels (b) and (d) are correct; both experiments (r230 and r251) lasted 75 seconds during the shearing stage.

Reviewer #2: Monica Barbery

This is a terrific manuscript focusing on flash heating in gouge using rotary shear experiments and IR temperature data across a wide range of sliding velocities and normal stresses. The manuscript is well-structured and concise, the methodology and modelling are well-conceived, and the figures are well-crafted. My recommendation is for minor revisions with only one major comment and a handful of minor comments detailed below. I look forward to seeing this manuscript published soon.

We thank Dr. Barbery for the positive evaluation of our manuscript. We appreciate the encouraging feedback and have addressed all comments in detail below. We hope that the revisions satisfactorily resolve the issues raised.

**Major comment:**
I think the discussion could benefit by briefly addressing energy budgets in these experiments, and more specifically the partitioning of energy going toward friction vs. fracture and comminution. I agree with many points in the discussion that at higher normal stresses (and sliding velocities), discrepancies between the predicted and measured temperatures could be related to either grain size reduction changing the asperity size and/or IR limitations. I also wonder if more energy is going to fracturing of grains and comminution in your higher normal stress experiments, and if the flash heating model predictions are overestimating the peak temperatures because they typically assume all the work is done by friction. This seems compatible with your measurements of enhanced grain size reductions at higher normal stresses and sliding velocities.

We thank Dr. Barbery for this thoughtful suggestion. We agree that considering the energy budget, particularly the partitioning between frictional work and grain comminution, may help assess whether additional energy sinks could contribute to the observed discrepancy between measured and predicted flash temperatures. To evaluate this, we estimated the surface fracture energy generated during grain comminution using particle size distributions of three representative experiments: (i) 2.5 MPa and 10 mm/s (r230), (ii) 5 MPa and 10 mm/s (r234), and (iii) 2.5 MPa and

50 mm/s (r228).

The surface fracture energy G was calculated as follows (Ma et al., 2006):

$$G_{surface} = \lambda G_c \frac{\Delta S}{A_{fault}} \qquad (1)$$

where $\Delta S$ is the new grain surface area created by comminution, $A_{fault}$ is the fault surface area (usually take 1 m²), $G_c$ is the specific fracture surface energy of the material (J/m²), $\lambda$ is the roughness factor (> 1, accounts for microscale roughness not captured by ideal shapes). The newly created surface area is

$$\Delta S = (1 - \varphi)h \sum_i \frac{6f_i}{d_i} \qquad (2)$$

where $h$ is the thickness of size reduction zone, $\varphi$ is the porosity, $f_i$ is the mass or volume fraction in size bin $i$, and $d_i$ their representative grain diameter. Using PSD data from 0.1 to 100 micron and assuming $h$ = 2 mmm, $\phi$ = 0.3, $G_c$ = 1 J/m², $\lambda$ = 1, we obtain r230 = ~4.19×10$^{-4}$ J/m², r234 = ~7.64×10$^{-4}$ J/m², r228 = ~5.99×10$^{-4}$ J/m². These estimates indicate that higher normal stress and slip velocity increase comminution-related fracture energy.

For comparison, we calculated the frictional work density FWD, representing the total work dissipated in the slip zone during slip):

$$FWD = \int_0^x \tau(x) dx \qquad (3)$$

with $x$ the total displacement and $\tau(x)$ the shear stress evolution with displacement $x$. We obtained r230 = ~1.22 MJ/m², r234 = ~2.44 MJ/m², r228 = ~1.22 MJ/m². Thus, the fracture energy is negligible relative to the frictional work—several orders of magnitude smaller. Although more energy is consumed by comminution at higher normal stress and slip velocity, the magnitude remains too small to meaningfully reduce the energy available for frictional heating or significantly affect flash temperature predictions. Therefore, while conceptually relevant, energy partitioning does not appear to be a controlling factor in our experiments.

Because these results do not alter our main interpretations, we have opted not to include this analysis in the manuscript; however, we provide it here in full for completeness and include one sentence in the main text to (Line 496-499):

*"The increase in fracture energy suggests a potential energy sink that is not converted into heat, although we found that this contribution is negligible compared to the frictional work."*

We hope that the response and revision satisfactorily addresses the reviewer's suggestion.

**Minor comments:**

Line 67 – Regarding the phrase "no visible dynamic weakening was observed", I'm a little unsure what this line references. I assume it refers to the measured surface temperatures not approaching flash heating breakdown temperatures of ~800 C? If this is the case, can you reword slightly for clarity? This disconnect between the maximum measured temperatures and the flash heating breakdown temperature could also be a function of the 75 mm resolution of the IR camera, which is roughly an order of magnitude larger than typical expectations for asperity sizes.

We thank Dr. Barbery for pointing out this unclarity. We have modified the sentence to clarify our reasoning: (Lines 67-71)
*"In addition, no evidence of dynamic weakening was detected, as the measured surface temperatures did not approach the weakening temperature inferred for granite (~800°C). This may also reflect wear processes that reduce local normal stresses and increase the true contact area during slip, thereby suppressing conditions favorable for flash heating."*

Figure 1 a & b: is the window 90° from the pore pressure inlet?

No, the window is 135° from the pore pressure inlet.

Table 1: I'm not sure exactly what number of flashes means. I presume this is the number of distinct contacts measured via IR in the window during experiments. Would that be correct? From lines 261-264 later, this seems to be the case. It could be helpful to define flashes here when introducing Table 1.

We thank Dr. Barbery for pointing out this unclarity. We have clarified the meaning of "flashes" in Table S1 to (Lines 168-170):
*"We define a flash as a maximum temperature measurement in a single frame that is larger (>50°C for dry, >4°C for wet) than the maximum temperatures in neighboring frames"*

Figure 5 a: There seems to be some evolution or hysteresis regarding the minimum measured temperatures in the 1 MPa experiment. Was there a temporary decrease in the sliding velocity or a hold around 400 mm of displacement?

We thank Dr. Barbery for noting this behavior. Indeed, the temporary decrease in the minimum measured temperature results from an unintended stop in shearing at 40 seconds due to a failure in the motor drive . Shearing was subsequently restarted at the same velocity for the remaining 30 seconds. We have added this clarification to the caption of Figure 5.

Figure 7 b: There are 3 spots with times close to 50s that are very hard to notice unless you look very closely, and 3 more I was unable to locate (I could only find 37 out of 40). A thin black line around these spots could perhaps help the reader see these spots more clearly. A scale bar for occurrence time (a,c) or temperature rise (b,d) could also be helpful here.

We thank Dr. Barbery for the suggestions. We have modified Figure 7 accordingly to increase the clarity as shown below.

[Figure]

*Figure 7. Spatial and temporal distribution of flashes within the main gouge zone with a minimum temperature rise ΔT (relative to the bulk maximum temperature) larger than 50°C and 4°C for the reference experiments of the room-dry (a and b; ID r230 with 40 flashes) and wet gouges (c and d; ID r251 with 13 flashes), respectively.*

*Tests performed at 2.5 MPa normal stress and 10 mm/s sliding velocity. Larger dots indicate later occurrence (a, c) or higher temperature rise (b, d).*

Figure 9 Line 414: Should this say "…sufficient spatial nor temporal resolution" instead of and/or? Assuming I followed correctly, the region above the red box has sufficient temporal resolution but not spatial resolution, and to the right of the red box you have sufficient spatial resolution but not sufficient temporal resolution, while the red region has neither the spatial nor temporal resolutions necessary to image contacts?

We thank Dr. Barbery for this clarification. You are correct—the red region represents conditions where neither the spatial nor temporal resolutions are sufficient to image grain contacts. We have revised the caption accordingly to reflect this.